

# Electronic excitations of the charged nitrogen-vacancy center in diamond obtained using time-independent variational density functional calculations

**Aleksei V. Ivanov[1*], Yorick L. A. Schmerwitz[2], Gianluca Levi[2] and Hannes Jónsson[3†]**

**1** Riverlane, St Andrews House, 59 St Andrews Street, Cambridge, CB2 3BZ, United Kingdom
**2** Science Institute of the University of Iceland, VR-III, 107 Reykjavík, Iceland
**3** Faculty of Physical Sciences, University of Iceland, VR-III, 107 Reykjavík, Iceland

★ alxvov@gmail.com , † hj@hi.is

## Abstract

Elucidation of the mechanism for optical spin initialization of point defects in solids in the context of quantum applications requires an accurate description of the excited electronic states involved. While variational density functional calculations have been successful in describing the ground state of a great variety of systems, doubts have been expressed in the literature regarding the ability of such calculations to describe electronic excitations of point defects. A direct orbital optimization method is used here to perform time-independent, variational density functional calculations of a prototypical defect, the negatively charged nitrogen-vacancy center in diamond. The calculations include up to 511 atoms subject to periodic boundary conditions and the excited state calculations require similar computational effort as ground state calculations. Contrary to some previous reports, the use of local and semilocal density functionals gives the correct ordering of the low-lying triplet and singlet states, namely $^3A_2 < {}^1E < {}^1A_1 < {}^3E$. Furthermore, the more advanced meta generalized gradient approximation functionals give results that are in remarkably good agreement with high-level, many-body calculations as well as available experimental estimates, even for the excited singlet state which is often referred to as having multireference character. The lowering of the energy in the triplet excited state as the atom coordinates are optimized in accordance with analytical forces is also close to the experimental estimate and the resulting zero-phonon line triplet excitation energy is underestimated by only 0.15 eV. The approach used here is found to be a promising tool for studying electronic excitations of point defects in, for example, systems relevant for quantum technologies.

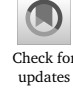

# 1   Introduction

The negatively charged nitrogen-vacancy center (NV$^-$ center) in diamond exhibits remarkable optical and magnetic properties, making it a promising candidate for quantum applications, including nanoscale sensing [1–4], quantum communication *via* single photon emission [5–7] and quantum bits (qubits) for quantum computing [8–12]. The applicability of the NV$^-$ center in quantum technologies derives from the possibility of generating a pure spin state with long coherence time through optical excitation. In order to optimally control the process and establish a theoretical approach that can guide the search of other systems for which a pure spin state can be prepared, accurate modelling of the electronic defect levels and corresponding excited states is desired.

Early theoretical studies of the NV$^-$ center in diamond using density functional theory (DFT) calculations with local and semilocal Kohn-Sham functionals [13–15] have led to contradictory results in that they do not agree on the ordering of the low-lying triplet and singlet defect states. In particular, there has been disagreement on which of two singlet states, $^1E$ or $^1A_1$, is lower in energy and whether or not both singlets lie between the two lowest triplet states. These conflicting results have contributed to a long-standing controversy regarding the electronic states involved in the optical spin-polarization cycle of the NV$^-$ center [16–18]. Only recently has this been resolved thanks to experimental observations and high-level many-body calculations [19–27] where the ordering

$$^3A_2 < {}^1E < {}^1A_1 < {}^3E$$

of the energy levels has been established. Spin initialization in the NV$^-$ center is realized through optical excitation from the triplet ground state ($^3A_2$) to an excited triplet state ($^3E$), the system then crossing over to an excited singlet state ($^1A_1$), followed by de-excitation to the lowest singlet state ($^1E$) and finally returning to the ground triplet state. Optical excitation occurs from both the $m_s = 0$ and $m_s = \pm 1$ sublevels of the triplet ground state, but electronic relaxation preferentially populates the $m_s = 0$ state. Therefore, the NV$^-$ center can be initialized in the $m_s = 0$ state after a few such optical cycles. The electrons that participate in the electronic transitions of the optical cycle occupy single-particle states between the valence and conduction bands and are thus localized at the defect.

As a result of the contradictory calculations mentioned above, there are several statements in the literature to the effect that the density functional approach cannot adequately describe the electronic structure of the NV$^-$ center in diamond and that calculations of the singlet electronic states require a higher level of theory because of multiconfigurational character.

Several different electronic structure methods are typically used for calculations of band gaps and defect levels in semiconductors for quantum technologies. A review is given in Ref. [28]. One of the most common methods is many-body perturbation theory based on the GW approximation [29], which uses the Green function formalism, together with the Bethe-Salpeter equation (BSE) [30,31]. However, it has been shown that the GW+BSE method does not provide a satisfactory description of the excited singlet $^1A_1$ state of the NV$^-$ center in diamond [16,26]. This has been attributed to the fact that the wave function of this state includes contributions of double excitations while GW+BSE only takes into account singly excited configurations [32]. Moreover, this approach involves large computational effort, too large for the supercells needed to accurately describe isolated defect states in semiconductors.

Time-dependent density functional theory (TDDFT) [33,34] is another commonly used method for calculating excited electronic states. Most TDDFT studies of the excited states of the NV$^-$ center in diamond have been carried out within the adiabatic and linear-response approximations and describe the system with a molecular cluster model where the surface atoms are saturated with bonds to hydrogen atoms [13,27,35,36]. Such finite models do not accurately describe the band structure and defect levels of semiconductors because of quantum confinement effects and the admixing of artificial surface states [28,32]. Recently, Galli and co-workers [20] have performed spin-flip TDDFT calculations of the NV$^-$ center in diamond using a supercell approach including periodic boundary conditions. Using the PBE functional as well as a hybrid functional that includes exact exchange, the right ordering of the states is obtained, but PBE gives too low energy for the excited singlet state, and the hybrid functional gives too high energy for both the singlet and triplet excited states, which was attributed to the lack of doubly excited configurations in TDDFT.

Spin defects in semiconductors have also been modelled using quantum embedding methods [21,22,24,37,38]. Here, the defect states are included in an active space described with a many-body effective Hamiltonian, and the interaction with the environment is taken into account through dielectric screening evaluated using DFT. This approach is found to estimate accurately the energy of the vertical excitations. However, quantum embedding calculations depend on the size of the active space, the method used to avoid double counting Coulomb interactions, and the procedure for obtaining the bulk screening, in addition to the choice of the energy functional [38]. Furthermore, since atomic forces are at present not available from quantum embedding calculations, the approach has so far relied on DFT calculations to estimate the effect of changes in the atomic coordinates in the excited states to obtain, for example, the zero-phonon line (ZPL) excitation energy.

While DFT is a ground state theory, various time-independent generalizations for calculating excited states exist [39–43]. In practical calculations, excited states can be found as higher energy stationary points on the surface describing how the energy varies as a function of the electronic degrees of freedom. Typically, these stationary solutions correspond to saddle points, making it necessary to employ an optimization method that avoids collapse to the ground state [44–52]. This approach [1] involves similar computational effort as ground state calculations and has proven useful for modelling excited states of both extended [46,53,54] and finite systems [45,47,51,55–60]. Since the excited states are obtained as stationary solutions of a set of single-particle equations, such as the Kohn-Sham equations [61], the orbitals are variationally optimized making it possible in principle to evaluate atomic forces analytically, thereby opening the possibility of performing atomic structure optimization and simulation of dynamics in the excited state [55,57,58,62]. Although only a single Slater determinant is optimized, complex potential energy surfaces for atomic motion have been shown to be described accurately, including avoided crossings and conical intersections (where TDDFT calcu-

---

[1]In the literature, variational density functional calculations of excited states are also referred to as delta self-consistent field ($\Delta$SCF) calculations.

lations are usually problematic), even for atomic configurations that are typically treated with a multiconfigurational approach [62–64]. For example, a conical intersection and avoided crossing in the ethylene molecule has been shown to be described well when symmetry breaking of the wave function is allowed for [63]. This is analogous to calculations of ground states that are inherently multiconfigurational (sometimes referred to as "strongly correlated") where symmetry breaking of the wave function gives improved estimate of the energy [65, 66], the stretched $H_2$ molecule being the classic example.

As mentioned above, DFT calculations of the excited states of the NV$^-$ center in diamond have given contradictory results. On the one hand, early calculations of Goss *et al.* [15] using a molecular cluster model and the local density approximation (LDA) functional [67] give the right ordering of the energy levels, $^3A_2 < {}^1E < {}^1A_1 < {}^3E$. On the other hand, the calculations of Delaney *et al.* [13] using larger molecular clusters and the Becke-Perdew (BP) exchange-correlation functional [68, 69] predict the $^1A_1$ singlet excited state to be higher in energy than the $^3E$ triplet excited state, and Gali *et al.* [14] reported calculations based on periodic boundary conditions and LDA where the singlet $^1A_1$ state is lower in energy than the singlet $^1E$ state, approximately isoenergetic with the $^3A_2$ triplet ground state. This disagreement between previous DFT calculations of the excited states of the NV$^-$ center in diamond is often taken as an indication of the inability of the method to describe multiconfigurational ("strongly correlated") states [16, 20, 22, 24]. However, single-determinant mean-field approaches have in many cases been shown to give quite good approximations to the energy of multiconfigurational systems, for example in bond-breaking processes and near avoided crossings, as well as for challenging molecules such as diradicals and the carbon dimer [63, 65, 66, 70, 71].

Here, the states of the NV$^-$ center in diamond are calculated using a recently developed direct orbital optimization method for a periodic supercell representation of the system and plane-wave basis set [45]. The functionals used include LDA, the generalized gradient approximation (GGA) PBE functional [72], and two meta-GGA functionals, TPSS [73] and r$^2$SCAN [74]. The calculations using any one of these functionals are found to give the correct ordering of the states, namely $^3A_2 < {}^1E < {}^1A_1 < {}^3E$, with the meta-GGA functionals providing vertical excitation energy values that are in remarkably close agreement with results of advanced quantum embedding calculations [21]. The relaxation energy in the triplet excited state and the ZPL energy of the optical transition from the triplet ground state are also found to be in good agreement with experimental estimates. The results presented here based on variational density functional calculations of the excited states demonstrate that this approach can indeed give accurate results even for open shell singlet states that are typically considered to have multireference character, and thereby provides a useful tool for modeling excited states of point defects in semiconductors with much smaller computational effort than the various higher level approaches.

## 2 Model and computational method

The NV$^-$ center in diamond consists of a substitutional nitrogen atom and a nearest-neighbour carbon vacancy and possesses trigonal C$_{3v}$ symmetry (see Figure 1). The low-lying triplet and singlet states can be described with three orbitals: a lower-energy $a_1$ orbital and a pair of higher-energy, degenerate $e$ orbitals, $e_x$ and $e_y$, that are localized on the carbon atoms around the vacancy. These orbitals are occupied by four electrons as shown in Figure 1. The $^3A_2$ triplet ground state can be represented with the $m_s = 1$ single Slater determinant $|e_y e_x\rangle$, hereafter denoted $^3\Phi_1$, where the $a_1$ orbital is doubly occupied and the $e_x$ and $e_y$ orbitals are singly occupied with spin up electrons. The $^3E$ triplet excited state is obtained by promotion of an electron from the $a_1$ orbital to one of the doubly degenerate $e$ orbitals and can be represented

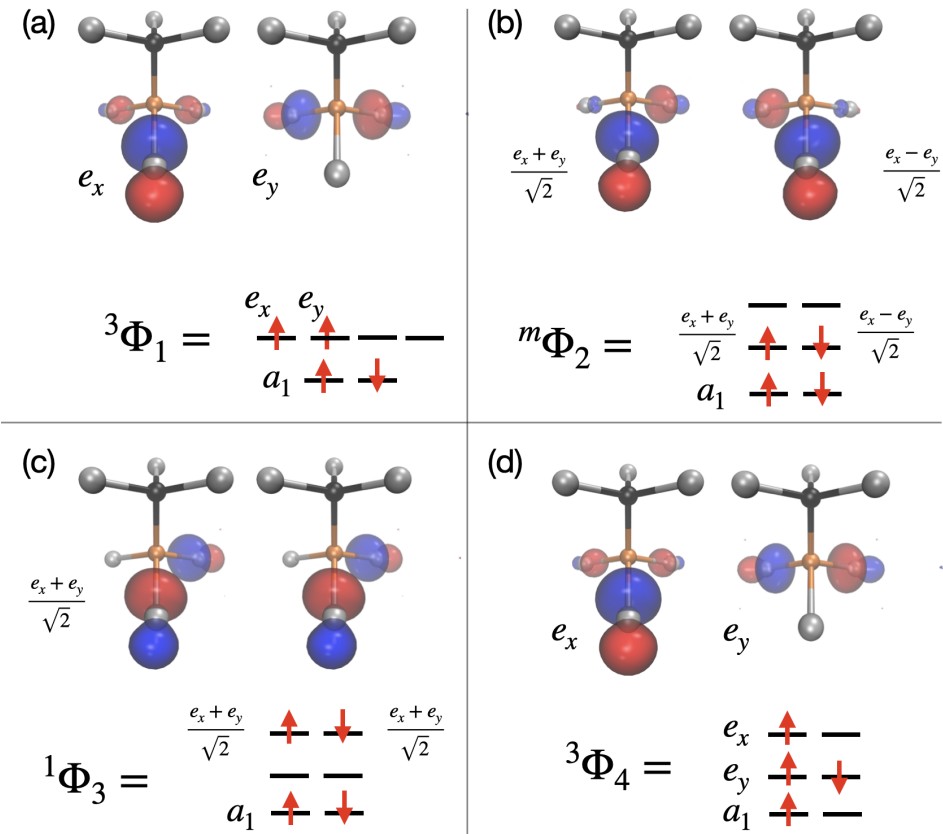

Figure 1: Atomic configuration of the NV⁻ center in diamond and representation of the orbitals corresponding to the defect levels lying within the band gap. C atoms: grey; N atom: dark grey; vacancy site: orange. The orbitals are obtained from PBE functional calculations of each one of the Slater determinants and are rendered at an isovalue of $0.25 \text{ Å}^{-3/2}$. The occupation of the orbitals in the determinants used to obtain the energy of the low-lying triplet and singlet states is also indicated. The ground and excited triplet states, $^3A_2$ and $^3E$, are calculated as the $m_s = 1$ single Slater determinants $^3\Phi_1$ and $^3\Phi_4$, respectively. The singlet states, $^1E$ and $^1A_1$, need to be represented by two or more Slater determinants and their energy is obtained from calculations of the ground triplet state determinant, $^3\Phi_1$, the spin-symmetry-broken determinant, $^m\Phi_2$, and the doubly excited spatial-symmetry-broken determinant, $^1\Phi_3$, according to Eqs. 5 and 6.

with the single determinant $|a_1e_x\rangle$, hereafter denoted $^3\Phi_4$.

The two singlet states, $^1E$ and $^1A_1$, have multideterminant character and need to be represented by two or more Slater determinants. The symmetry-adapted wave functions of these states are [15, 75]

$$\Psi\left(^1E\right) = \begin{cases} \left(|e_x\overline{e_y}\rangle + |e_y\overline{e_x}\rangle\right)/\sqrt{2}, \\ \left(|e_x\overline{e_x}\rangle - |e_y\overline{e_y}\rangle\right)/\sqrt{2}, \end{cases} \tag{1}$$

$$\Psi\left(^1A_1\right) = \left(|e_x\overline{e_x}\rangle + |e_y\overline{e_y}\rangle\right)/\sqrt{2}. \tag{2}$$

By introducing the following linear combinations of the $e_x$ and $e_y$ orbitals

$$e_- = \frac{e_x - e_y}{\sqrt{2}}, \tag{3}$$

$$e_+ = \frac{e_x + e_y}{\sqrt{2}}, \tag{4}$$

the energy of the multideterminant singlet states can be calculated from the energy of single determinants using the formulas (see Appendix A for a derivation)

$$\mathcal{E}[^1E] = 2\mathcal{E}[^m\Phi_2] - \mathcal{E}[^3\Phi_1], \tag{5}$$

$$\mathcal{E}[^1A_1] = \mathcal{E}[^3\Phi_1] + 2(\mathcal{E}[^1\Phi_3] - \mathcal{E}[^m\Phi_2]), \tag{6}$$

where $^m\Phi_2 = |e_-\overline{e_+}\rangle$ is a determinant with broken spin symmetry and $^1\Phi_3 = |e_+\overline{e_+}\rangle$ is a doubly excited determinant with broken spatial symmetry (see Figure 1). Eq. (5) represents spin-purification [76,77].

Calculations are carried out using the following density functionals: LDA [67], PBE [72], TPSS [73], and r$^2$SCAN [74]. The orbitals are represented with a plane-wave basis set and the projector augumented wave method [78]. The plane-wave basis corresponds to a 600 eV kinetic energy cutoff. First, the lattice parameters of a 512-atom supercell of diamond are optimized using the PBE functional. Then, the NV$^-$ defect center is introduced and the resulting structure optimized in the ground triplet state represented by the $^3\Phi_1$ determinant for each of the chosen functionals until the largest atomic force is below 0.01 eV/Å. The energy of vertical excitations for both singlet and triplet states is obtained at the geometry optimized in the triplet ground state. The excited state determinants, $^1\Phi_3$ and $^3\Phi_4$, correspond to saddle points on the electronic energy surface. All determinants have been calculated using the direct orbital optimization method presented in Ref. [45] which makes use of a limited-memory version of the symmetric rank-one (L-SR1) quasi-Newton algorithm [47] to assist the convergence on the saddle points. Calculations use integer occupation numbers and no symmetry constraints are enforced. The ZPL energy for the optical transition between the two triplet states is obtained by optimizing the atomic structure in the excited triplet state and then evaluating the energy difference with respect to the relaxed ground state. All calculations are performed with the GPAW [79], Libxc [80] and ASE [81] software. Visualization of orbitals has been carried out with the VMD software [82]. In order to ensure the 511-atom supercell (with the vacancy) is large enough, calculations were also carried out with a smaller cell containing 215 atoms, and the results were found to differ by at most 5 meV (see Tables 2, and 3 in the Appendix). Given this small difference, it is appropriate to compare the results obtained here with a 511-atom supercell with the previous periodic calculations where a 215-atom supercell was used.

## 3 Results

Figure 2 shows results obtained here with the LDA functional (leftmost column). The right ordering of the energy levels is obtained, $^3A_2 < {}^1E < {}^1A_1 < {}^3E$. The figure also shows a comparison with two previously reported LDA calculations (second and third column), one based on periodic boundary conditions as in the present study [14], and another for a molecular cluster [15]. The molecular cluster LDA calculations of Goss et al. [15] also get the right ordering of the energy levels, whereas Gali et al. [14] report a singlet $^1A_1$ energy close to that of the $^3A_2$ triplet ground state, which is in strong contradiction with high level calculations as well as experimental measurements. It is unclear what the reason for this discrepancy is. The figure also shows results obtained with the BP gradient dependent functional [13] (fourth

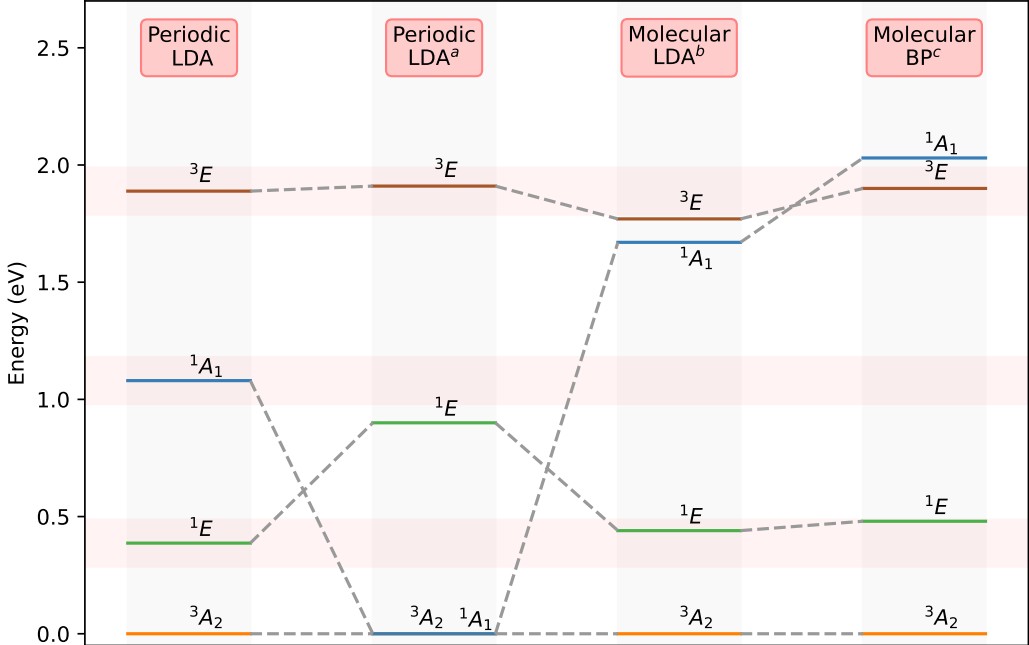

Figure 2: Energy of vertical excitations of the NV⁻ center in diamond from the triplet ground state. The leftmost column shows results obtained here from variational calculations with the LDA functional. The red horizontal shading indicates a range of ±0.1 eV around those values. The ordering of the states, $^3A_2 < {}^1E < {}^1A_1 < {}^3E$, is in agreement with the best estimates. For comparison, published results of LDA calculations [14], also using periodic boundary conditions, are shown in the next column. There, the ordering of states is different for some unknown reason. The third column shows results of molecular cluster LDA calculations [15]. The last column shows results of calculations [13] using the gradient-dependent BP functional, where the ordering of the excited triplet and excited singlet states is reversed. $^a$Ref. [14], $^b$Ref. [15], $^c$Ref. [13].

column) and a cluster model. There, the ordering of the excited triplet and excited singlet states is reversed, $^3E < {}^1A_1$.

Figure 3 shows results obtained here with GGA and meta-GGA functionals, in addition to the LDA results, as well as a comparison with results obtained using various many-body calculations. The numerical values are listed in Table 2 in the Appendix. The correct ordering of the energy levels is obtained for all of the functionals. The largest absolute changes in excitation energy with respect to the density functional are obtained for the excited singlet state, $^1A_1$, but the relative changes of the two singlet states are similar. The excited triplet state, which is the only state apart from the ground state that can be described using a single determinant, is affected less by the choice of functional. The values of the excitation energy increases as the complexity of the functional increases, in the order LDA < GGA < meta-GGA, with the recently developed r$^2$SCAN functional also giving slightly larger values for the energy of the vertical excitations than the TPSS functional.

The most accurate values are believed to be the results of Ma *et al.* [21] shown in the fifth column from the left in Figure 3. These are obtained using quantum embedding calculations beyond the random phase approximation (from now on referred to as 'beyond-RPA', but labeled bRPA in the figure) and include explicit exchange-correlation effects of the environment on the defect energy levels. Column six shows results of periodic quantum embedding calculations where the screened Coulomb interactions are evaluated using a constrained random phase



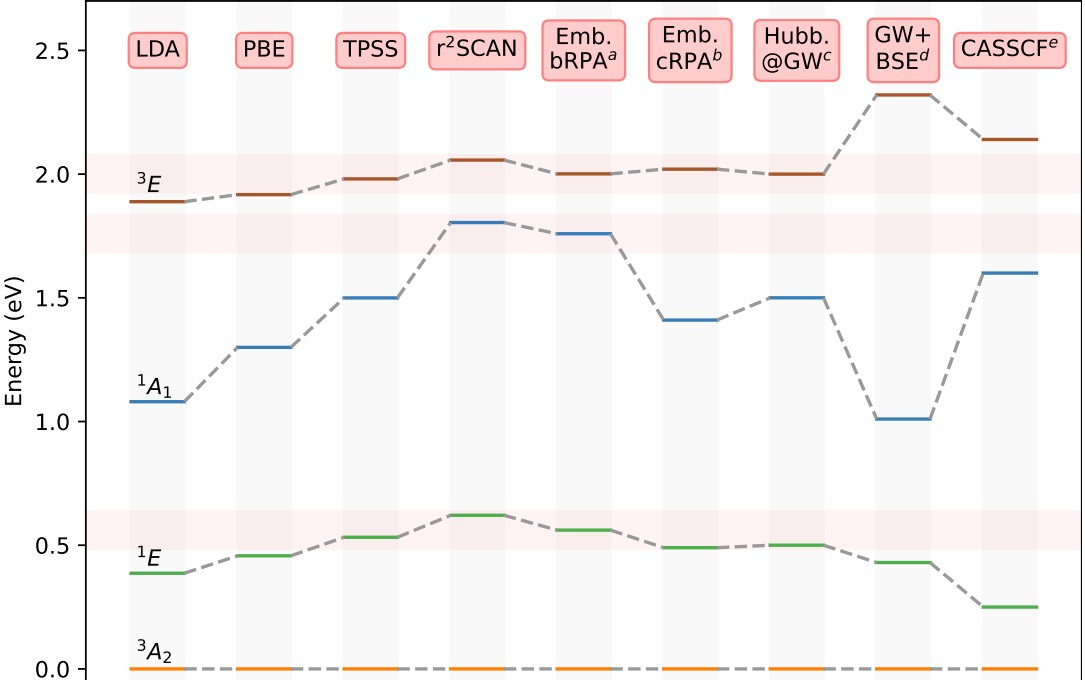

Figure 3: Energy of vertical excitations relative to the triplet ground state of the NV⁻ center in diamond obtained with variational calculations using different local and semilocal density functional approximations, and comparison with results of previous calculations based on many-body approaches: periodic quantum embedding beyond the random phase approximation, beyond-RPA (bRPA) [21], constrained RPA (cRPA) [24], extended Hubbard model fitted to GW calculations [16], periodic GW+BSE [26], and molecular cluster complete active space self-consistent field (CASSCF) [23] calculations. The red horizontal shadings span ±0.075 eV around the values obtained with the beyond-RPA quantum embedding results [21], which are taken to give the best theoretical estimates. The $r^2$SCAN functional gives results that are remarkably close, the largest deviation being below 0.06 eV. All four density functionals used here give the correct ordering of the energy levels of the electronic states. [a]Ref. [21], [b]Ref. [24], [c]Ref. [16], [d]Ref. [26] [e]Ref. [23].

approximation (cRPA), neglecting exchange-correlation effects [24]. There, the energy of the excited singlet state is significantly lower than in the beyond-RPA calculation.

The remaining columns in figure 3 show results of an extended Hubbard model fitted to GW calculations (Hubb.@GW) [16], periodic GW+BSE calculations [26], and recent molecular cluster complete active space self-consistent field (CASSCF) calculations with state averaging [23]. In all cases the ordering of the states is the same, but the two singlet states differ the most, consistent with the fact that they have multireference character. The GW+BSE calculation clearly gives a too large value for the excited triplet state.

It is clear from the results shown in figure 3 that the $r^2$SCAN functional provides the best results of the four functionals used in the present study. Remarkably close agreement is obtained with the most accurate results coming from the beyond-RPA calculations. The deviation of the values obtained with $r^2$SCAN from those of the beyond-RPA values is below 0.06 eV for all the excited states (see Table 1). We note that an earlier version of the SCAN functional [83] gives quite similar results (see Table 2 in the Appendix). Calculations using the TPSS functional also provide results in quite good agreement with the beyond-RPA calculations, with the largest difference being in the excited singlet state, $^1A_1$, where the vertical excitation energy is

Table 1: Calculated energy of vertical excitations (in eV) between states of the NV⁻ center in diamond obtained with the $r^2$SCAN functional, and taken from reported theoretical best estimates using a beyond-RPA method [21] as well as the difference between the two. The $r^2$SCAN results are remarkably close to the beyond-RPA results.

| | $^3A_2 \leftrightarrow {}^3E$ | $^3A_2 \leftrightarrow {}^1A_1$ | $^3A_2 \leftrightarrow {}^1E$ | $^1E \leftrightarrow {}^1A_1$ | $^1A_1 \leftrightarrow {}^3E$ |
|---|---|---|---|---|---|
| $r^2$SCAN | 2.057 | 1.804 | 0.621 | 1.183 | 0.253 |
| beyond-RPA (Ref. [21]) | 2.001 | 1.759 | 0.561 | 1.198 | 0.243 |
| Difference | 0.056 | 0.045 | 0.060 | -0.015 | 0.010 |

underestimated by ∼0.25 eV. A similar deviation is obtained in the Hubbard-model and cRPA many-body calculations. The largest deviations in the LDA and PBE calculations are also in the energy of excitation to the $^1A_1$ state, which is underestimated with respect to the beyond-RPA results by ∼0.70 and ∼0.45 eV, respectively. The excitation energy for the $^1E$ singlet ground state and $^3E$ triplet excited state are predicted within ±0.1 eV from the values of the beyond-RPA reference calculations for all the density functionals used here. In comparison, the more computationally intensive GW+BSE calculations of ref. [26] give an accuracy comparable to LDA for the $^1A_1$ state and have relatively large deviation of 0.3 eV with respect to the beyond-RPA result for the $^3E$ triplet excited state. The CASSCF calculations of ref. [23] also have larger errors for the $^3E$ and $^1E$ states than all the density functionals employed here, including LDA.

Figure 4 compares the ZPL energy for the transition between triplet states, $^3A_2 \leftrightarrow {}^3E$, calculated using the various density functionals (values reported in Table 2 in the Appendix) with the experimental ZPL energy, which is 1.945 eV [84]. Again, $r^2$SCAN gives the best results with a deviation of -0.15 eV with respect to the experimental value. This is only slightly larger than the error of previous periodic calculations with the screened hybrid functional Heyd-Scuseria-Ernzerhof (HSE) [86, 87], where it was found that the ZPL energy is overestimated by 0.1 eV compared to experiment. Calculations using the TPSS functional underestimate the ZPL energy of the triplet transition by 0.2 eV, while calculations using the PBE and LDA functionals give a deviation of 0.27 eV, similar to previous periodic calculations using local and semilocal functionals [86]. All functionals reproduce well the experimentally determined energy lowering after excitation to the $^3E$ state, 0.235 eV [84], to within 0.035 eV.

Figure 4 also shows the experimentally deduced excitation energy that includes the lowering of the energy after excitation to the singlet states. The ZPL energy for the transition between singlets, $^1E \leftrightarrow {}^1A_1$, has been determined to be 1.19 eV [85]. The energy of the singlet states with respect to the triplet ground state can be deduced from the measured ionization energy of the singlet ground state of 1.91-2.25 eV obtained from recent photoluminescence measurements [19]. Theoretical estimates of the lowering of the energy in the singlet states have recently been presented by Jin *et al.* [20] who report 0.06-0.1 eV and 0.02 eV for the $^1E$ and $^1A_1$ states, respectively, on the basis of spin-flip TDDFT calculations with the PBE functional. This suggests that the ZPL singlet energy should differ from the $^1E \leftrightarrow {}^1A_1$ vertical excitation energy by at most 0.08 eV. From this, it can be deduced that $r^2$SCAN predicts the ZPL energy for the singlet transition with an accuracy of 0.03-0.07 eV with respect to the experimentally deduced value, while the other functionals give too small ZPL energy because the energy of the $^1A_1$ state is underestimated. The $r^2$SCAN functional however underestimates the energy difference of 0.24 eV between the $^1A_1$ and $^3E$ excited states. There, the TPSS functional appears to give a better estimate of ∼0.2 eV, in good agreement with the experimentally deduced value.

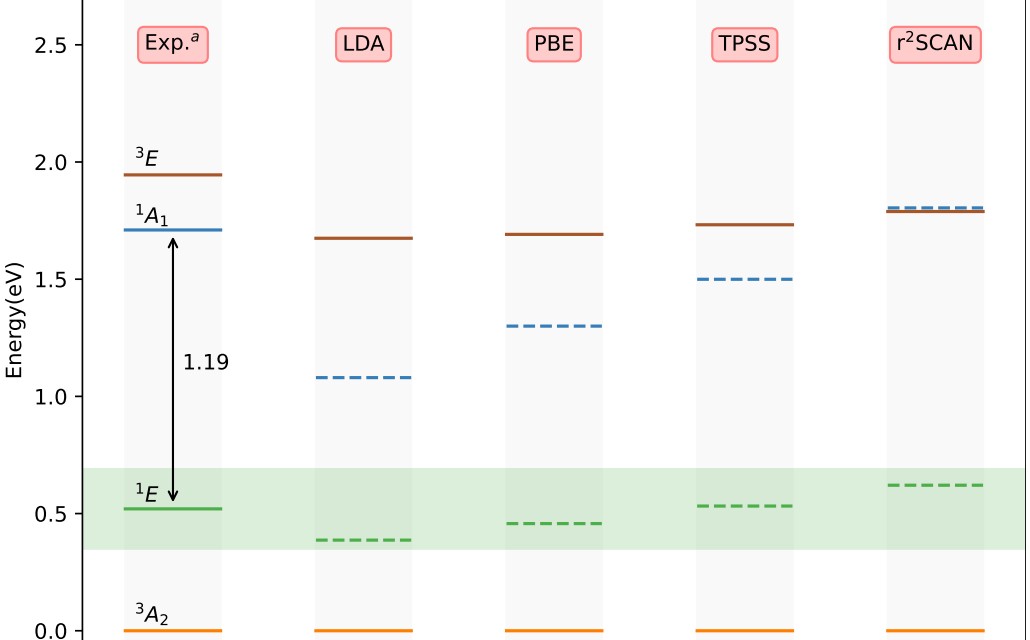

Figure 4: Energy of zero-phonon line (ZPL) excitations (solid lines) of the NV⁻ center in diamond obtained experimentally and from variational calculations using the four density functionals, as well as the energy of vertical excitations (dashed lines, same values as in Figure 3) to the singlet states where energy lowering due to changes in atomic coordinates have not been included. The green horizontal shading represents the uncertainty in the experimental value of the ionization energy of the singlet ground state [19]. The results obtained with the $r^2$SCAN underestimate the experimental ZPL triplet energy by only 0.15 eV, while LDA has the largest error (0.27 eV). The energy lowering due to relaxation of atomic coordinates in the singlet states has been estimated recently using spin-flip TDDFT calculations giving 0.06-0.1 eV for $^1E$ and 0.02 eV for $^1A_1$ [20]. Applying these corrections to the $r^2$SCAN values for the singlets gives ZPL energy of the singlet transition close to the experimental estimate, 1.19 eV, but underestimates the difference between the $^1A_1$ and $^3E$ excited states. The results using the TPSS functional provide a more accurate value of the $^1A_1 - {}^3E$ energy difference. $^a$Ref. [19, 84, 85].

## 4  Discussion and conclusions

The variational calculations presented here using several density functionals, ranging from LDA to meta-GGAs, show that the ordering of the four lowest-energy states of the NV⁻ center in diamond is actually predicted correctly at this level of theory, contrary to some earlier reports. The energy of vertical excitations obtained with the meta-GGA functionals are, furthermore, in close agreement with the results of accurate but much more computationally intensive many-body beyond-RPA quantum embedding calculations [21]. The best results are obtained with the $r^2$SCAN functional, with deviations from the beyond-RPA values of only ∼3% and ∼1% for the optical transitions between the triplet and singlet states, $^3A_2 \leftrightarrow {}^3E$ and $^1E \leftrightarrow {}^1A_1$, respectively.

The calculations presented here are based on a recently developed direct orbital optimization approach and a plane-wave basis set for a large supercell with up to 511 atoms subject to periodic boundary conditions in order to ensure convergence with respect to size and without introducing artifacts due to finite size and truncated surfaces. The single-determinant wave

functions for the singlet states allow for symmetry breaking, as this is known to be an important feature for obtaining accurate estimates of the energy of multiconfigurational ("strongly correlated") systems within a mean-field approximation [63, 65, 66]. We note that the commonly used self-consistent field optimization of orbitals typically converges on a symmetric solution if it is started from a symmetric initial wave function.

Since the calculations are variational, the atomic forces in the excited states can in principle be evaluated analytically. The minimization of the energy in the $^3E$ triplet excited state gives an estimate of the ZPL energy for the triplet transition in good agreement with experimental estimates, with r$^2$SCAN underestimating the experimental value by $\sim$0.15 eV. This error is similar to the one previously found for the much more elaborate and computationally intensive HSE hybrid functional [86, 87]. Optimization of atomic coordinates and simulation of the atomic dynamics in the singlet states $^1E$ and $^1A_1$ is, however, not straightforward because it involves determination of the atomic forces for multiple single-determinant solutions as indicated in Eqs. 5 and 6. Moreover, since the description of the $^1A_1$ state involves a single determinant that breaks the spatial symmetry of the wave function, optimization of atomic coordinates in this state may lead to artificial symmetry breaking of the atomic configuration of the NV$^-$ center. This issue can be alleviated by introducing a basis of complex orbitals expressed as a linear combination of the real $e_x$ and $e_y$ orbitals, where the $^1A_1$ state can be described with a single-determinant solution that does not break the spatial symmetry [88].

The values of the excitation energy obtained with the LDA and PBE functionals are found to be underestimates with respect to the beyond-RPA results, especially for the excited singlet state, $^1A_1$. This underestimation can be a consequence of the self-interaction error inherent in calculations with local and semilocal Kohn-Sham functionals, which can be different for the different states and thereby affect the excitation energy. An explicit self-interaction correction can be applied using the orbital based approach proposed by Perdew and Zunger [89]. Such a correction applied to the PBE functional has, for example, been found to significantly improve the calculated energy of excitations of the ethylene molecule [63]. This approach might also give improved results for the NV$^-$ center in diamond, but the calculations are more involved and computationally demanding as they require an additional optimization of the orbitals due to the lack of unitary invariance of the corrected functional [45, 90, 91].

Variational density functional calculations where excited states are obtained as stationary single-determinant solutions have in some articles in the literature been described as inadequate for describing electronic excitations of quantum defects. The results of the calculations presented here show instead that such an approach can provide accurate energetics for the electronic states involved in the optical spin initialization in the prototypical NV$^-$ center in diamond. This methodology is, therefore, expected to be a useful tool for characterizing electronic excitations of other point defects in materials of interest for quantum applications.

## Acknowledgment

We thank Elvar Ö. Jónsson and Sergei Egorov for helpful discussions.

**Funding information** This work was funded by the Icelandic Research Fund (grant nos. 174582, 217734) and by the Research Fund of the University of Iceland.

# A Relation between multideterminant states and single determinants

Using Eqs. 3 and 4, the single Slater determinants ${}^m\Phi_2$ and ${}^1\Phi_3$ can be expanded as

$$
{}^m\Phi_2 = |e_-\overline{e_+}\rangle = \left(|e_x\overline{e_x}\rangle - |e_y\overline{e_y}\rangle\right)/2 + \left(|e_x\overline{e_y}\rangle - |e_y\overline{e_x}\rangle\right)/2
$$
$$
= \frac{1}{\sqrt{2}}\left(\Psi\left({}^1E\right) + \Psi\left({}^3A_2\right)\right), \tag{A.1}
$$

$$
{}^1\Phi_3 = |e_+\overline{e_+}\rangle = \left(|e_x\overline{e_x}\rangle + |e_y\overline{e_y}\rangle\right)/2 + \left(|e_x\overline{e_y}\rangle + |e_y\overline{e_x}\rangle\right)/2
$$
$$
= \frac{1}{\sqrt{2}}\left(\Psi\left({}^1A_1\right) + \Psi\left({}^1E\right)\right), \tag{A.2}
$$

where it has been used that the many-body singlet wave functions are given by the expressions in Eqs. 1 and 2 and that $\left(|e_x\overline{e_y}\rangle - |e_y\overline{e_x}\rangle\right)/2$ is the $m_s = 0$ triplet ground state wave function. Eq. A.1 shows that the single determinant ${}^m\Phi_2$ is a mixture of the multideterminant ground singlet and triplet wave functions, $\Psi\left({}^1E\right)$ and $\Psi\left({}^3A_2\right)$, which leads to spin-symmetry breaking. Eq. A.2 shows that the single determinant ${}^1\Phi_3$ is a mixture of the multideterminant ground and excited singlet wave functions, $\Psi\left({}^1A_1\right)$ and $\Psi\left({}^1E\right)$, which leads to spatial-symmetry breaking.

Since the Hamiltonian matrix elements between wave functions of different spin or spatial symmetry are zero, the energy of the single determinants ${}^m\Phi_2$ and ${}^1\Phi_3$ can be expressed as

$$
\mathcal{E}\left[{}^m\Phi_2\right] = \left(\mathcal{E}\left[{}^1E\right] + \mathcal{E}\left[{}^3A_2\right]\right)/2, \tag{A.3}
$$
$$
\mathcal{E}\left[{}^1\Phi_3\right] = \left(\mathcal{E}\left[{}^1E\right] + \mathcal{E}\left[{}^1A_1\right]\right)/2. \tag{A.4}
$$

Therefore, in the absence of orbital relaxation, the energy of the multideterminant ground and excited singlet states is given in terms of the energy of the single determinants ${}^m\Phi_2$, ${}^3\Phi_1$ and ${}^1\Phi_3$ by

$$
\mathcal{E}\left[{}^1E\right] = 2\mathcal{E}\left[{}^m\Phi_2\right] - \mathcal{E}\left[{}^3A_2\right] = 2\mathcal{E}\left[{}^m\Phi_2\right] - \mathcal{E}\left[{}^3\Phi_1\right], \tag{A.5}
$$
$$
\mathcal{E}\left[{}^1A_1\right] = 2\mathcal{E}\left[{}^1\Phi_3\right] - \mathcal{E}\left[{}^1E\right] = \mathcal{E}\left[{}^3\Phi_1\right] + 2(\mathcal{E}\left[{}^1\Phi_3\right] - \mathcal{E}\left[{}^m\Phi_2\right]), \tag{A.6}
$$

where the energy of the ground triplet state is evaluated from the $m_s = 1$ single determinant ${}^3\Phi_1$ instead of the $m_s = 0$ state. This is a good approximation since the splitting between the $m_s = 0$ and $m_s = \pm 1$ levels in the triplet ground state is $\sim 2.88$ GHz ($\sim 10^{-5}$ eV) as determined by electron paramagnetic resonance measurements [10, 92].

## B  Vertical excitation and zero-phonon line

Table 2: Vertical excitation energy (in eV) of the NV⁻ center in diamond obtained from variational calculations using various local and semilocal density functionals. For the singlet states, the energy has been calculated according to Eqs. 5 and 6, taking into account the multideterminant character of these states. The values in parentheses correspond to the calculated zero-phonon line (ZPL) energy of the optical transition between the triplet states. $^a$Ref. [21].

|  | $^3A_2 \leftrightarrow {}^3E$ (ZPL) | $^3A_2 \leftrightarrow {}^1A_1$ | $^3A_2 \leftrightarrow {}^1E$ | $^1E \leftrightarrow {}^1A_1$ | $^1A_1 \leftrightarrow {}^3E$ |
|---|---|---|---|---|---|
| LDA | 1.889 (1.675) | 1.080 | 0.387 | 0.693 | 0.809 |
| PBE | 1.917 (1.691) | 1.300 | 0.457 | 0.843 | 0.617 |
| TPSS | 1.981 (1.732) | 1.500 | 0.532 | 0.968 | 0.481 |
| SCAN | 2.069 (1.832) | 1.980 | 0.654 | 1.326 | 0.089 |
| r$^2$SCAN | 2.057 (1.789) | 1.804 | 0.621 | 1.183 | 0.253 |
| beyond-RPA$^a$ | 2.001 (  -  ) | 1.759 | 0.561 | 1.198 | 0.243 |

## C  Smaller cell calculations

Table 3: Vertical excitation energy (in eV) obtained from variational calculations with the PBE functional on a 215-atom supercell of the NV⁻ center in diamond. The atomic structure of this supercell corresponds to the structure optimized with PBE in the ground triplet state in Ref. [21]. A kinetic energy cutoff of 600 eV is used. The values of excitation energy differ by at most 5 meV from the values obtained using PBE with the larger 511-atom supercell shown in Table 2.

|  | $^3A_2 \leftrightarrow {}^3E$ | $^3A_2 \leftrightarrow {}^1A_1$ | $^3A_2 \leftrightarrow {}^1E$ |
|---|---|---|---|
| PBE | 1.921 | 1.295 | 0.459 |

## D  Supplemental information

Data related to the results presented here is available at Zenodo [93].

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
