# Peer review of "Electronic excitations of the charged nitrogen-vacancy center in diamond obtained using time-independent variational density functional calculations"

_SciPost Physics, doi:SciPost Phys. 15, 009 (2023)_

## Round 1 · Referee Report · Anonymous (Referee 1) · 2023-4-25

Report

In this manuscript A. V. Ivanov and co-workers applied time-dependent variational density functional theory to describe the electronic excitations of a prototypical defect, the negatively charged nitrogen-vacancy center in diamond. They used 4 local and semi-local density functionals and compared their results with previous density functional based calculations, high-level many-body calculations and available experimental data. They report that their method is more robust and predictive than DFT based calculations, since a correct ordering of the low-lying triplet and singlet states is obtained, in qualitative and quantitative agreement with the aforementioned high-level methods.

The resolution of the correct order controversy raised by previous studies is the major advance of the paper.

The paper is well written, interesting and it deserves to be published, but in my opinion it should go to Scipost core, as it’s not a breakthrough. Here are my comments and remarks.

1) The authors used a 512 atoms supercell, how did the authors check if the supercell is “big enough” ? 2) It seems to me that people in the literature claim that the correct order of the low lying triplet and singlet states is not recovered because of the fact that the methods used are not able to describe multi-configurational states. However, as far as I understand, the Delta SCF method is also based on a single determinant state. So then what is the strength of this method? Is it the fact that the orbitals are variationally optimised and then the forces can, in principle, be calculated analytically? 3) Did the authors rely on this point to optimize the triplet excited states and calculate the ZPL? 4) From Fig. 2 and the discussion, it seems that the A1 state is most affected by the type of calculation, do the authors have an opinion on this? 5) The most complex functional r2SCAN gives vertical excitations in very good agreement with quantum embedding calculations beyond the random phase approximation (called Emb. bRPA). What makes this method so similar to the Emb. bRPA? 6) What geometry did the authors impose for the evaluation of the vertical excitation of the singlet states? They say that the energy decrease due to changes in atomic coordinates was not evaluated. 7) Would it be possible to provide a pictorial view of the electronic energy surfaces, with saddle points, position of triplet and singlet state geometries and the ZPL? I think that even a simple sketch would also help to follow the discussion.

Attachment

  • validity: -
  • significance: -
  • originality: -
  • clarity: -
  • formatting: -
  • grammar: -

Author:  Hannes Jonsson  on 2023-05-04  [id 3640]

(in reply to Report 1 on 2023-04-25)
Category:
answer to question

We agree that our manuscript does not present a breakthrough in terms of improved understanding of the NV defect in diamond, but it opens up a new pathway for computational studies of excited defect states in solids in that the approach we present can give results in close agreement with higher level many-body approaches that are much more computationally intensive. We show that, contrary to common belief and several published articles, calculations based on DFT can give accurate results if carried out as described in the manuscript. As the computational effort is moderate, our approach can be used in routine calculations to help understand measurements carried out on other systems and predict properties of new ones. Furthermore, the calculations give atomic forces because they are variational and this makes efficient optimization of atomic coordinates in excited states possible and even simulations of the dynamics. Below is our response to the questions raised in the report:

  1. We also carried out calculations on a smaller system of 215 atoms to test whether the larger one we report on is large enough. We note that previous studies, such as the beyond-RPA calculations, were carried out for the smaller supercell. The changes in the energy values are within 5 meV (now presented in Table 3). The following has been added at the end of section 2:  "In order to ensure the 511-atom supercell (with the vacancy) is large enough, calculations were also carried out with a smaller cell containing 215 atoms, and the results were found to differ by at most 5 meV (see Tables 2, and 3 in the Appendix). Given this small difference, it is appropriate to compare the results obtained here with a 511-atom supercell with the previous periodic calculations where a 215-atom supercell was used." 

  2. In our calculations, the variationally optimized single determinant solutions are used to construct multideterminant estimates of the energy levels of the singlet states, see equations (5) and (6). We also make sure that the calculations allow for symmetry breaking since this can account for multiconfigurational character (see equations (7) and (8)). SCF calculations typically converge on a symmetric solution and fail to find the broken symmetry solution if they are started from a symmetric initial guess.  We have added the following sentence at the end of the second paragraph of section 4: "We note that the commonly used self-consistent field optimization of orbitals typically converges on a symmetric solution if it is started from a symmetric initial wave function."

  3. We indeed made use of the variational nature of the calculated solutions to obtain the atomic forces in the triplet excited state and optimized the atomic coordinates to evaluate the ZPL.

  4. The energy of both singlet states with respect to the triplet ground state is in fact affected strongly by the choice of the functional, while the triplet excited state, which can be described using a single determinant, is less affected. Most likely this sensitivity of the singlet states is related to the multiconfigurational character. We have rephrased a bit the discussion in the second paragraph of the Results section to: "The correct ordering of the energy levels is obtained for all of the functionals. The largest absolute changes in excitation energy with respect to the density functional are obtained for the excited singlet state, 1A1, but the relative changes of the two singlet states are similar. The excited triplet state, which is the only state apart from the ground state that can be described using a single determinant, is affected less by the choice of functional.”

  5. All sound approaches converge to the same, correct answer if carried out to completion, irrespective of how they reach that limit. While the meta-GGA functional and the beyond-RPA approaches are very different, they both represent the highest level of theory within their respective category. It is therefore not surprising that they give similar results. We do not, however, see a mathematical connection between the two.

  6. The vertical excitation energy of the singlet states is calculated for the geometry optimized in the triplet ground state. To clarify this, we have modified slightly a sentence in the last paragraph of section 2 to read "The energy of vertical excitations for both singlet and triplet states is obtained at the geometry optimized in the triplet ground state".

  7. It is extremely challenging to give a pictorial view of the variation of the energy as a function of all the different types of degrees of freedom involved, i.e. both electronic and atom coordinates. But, each of these can be visualized to some extent separately. We point to another manuscript from the group where the saddle points on the electronic energy surface corresponding to excited electronic states of the H2 molecule are visualized, see figures 1 and 3 in https://arxiv.org/abs/2302.05912. We will in a future publication take on this challenge for defect states in solids.

---

## Round 1 · Referee Report · Anonymous (Referee 2) · 2023-4-27

Report

The manuscript is a nice application of the authors' recently developed method(https://doi.org/10.1021/acs.jctc.1c00157) for calculating excited states using density functional theory. Using their recent developments, they are able to show that they can obtain the same ordering as many-body calculations (and experimental estimates) for the low-lying triplet and singlet states of the NV- center in diamond, namely 3A2 < 1E < 1A1 < 3E.
The manuscript is well written and shows results for their promising new method for calculating excited states and electron transfer calculations. I found the section on the "Relation between multi-determinant states and single determinants" particularly interesting. Their presentation seems to follow traditional quantum chemistry logic and spin-contamination/symmetry breaking. Obviously, for some solid state systems, e.g., iron-oxides, spin-contamination is excepted, whereas for small molecules such symmetry breaking is thought to be bad, mostly due to our experience of using UHF theory with many organic molecules. However, I'm not as sure I understand the rules of the road for spin contamination anymore, since it has been recently suggested that spin-symmetry breaking in the simple C2 molecule gives the correct energetics of the C2 singlet and low-lying triplet, especially when the SCAN meta-GGA is used (https://pubs.acs.org/doi/10.1021/acs.jpca.2c07590). I guess my question about their excited state results for NV- center, are they seeing a similar result in their calculations? Where SCAN is producing the correct energetics for low-lying excited states.....or do the authors think the results of these studies are unrelated.

Requested changes

No changes are required

  • validity: top
  • significance: high
  • originality: high
  • clarity: top
  • formatting: excellent
  • grammar: excellent

Author:  Hannes Jonsson  on 2023-05-04  [id 3641]

(in reply to Report 2 on 2023-04-27)

We thank the referee for mentioning this article (Perdew et al. "Symmetry Breaking with the SCAN Density Functional Describes Strong Correlation in the Singlet Carbon Dimer" J. Phys. Chem. A 127, 389 (2023), reference 65 in our manuscript). Symmetry breaking is indeed also very important in our NV-diamond calculations, not only with the r2SCAN functional but also the other functionals. With the symmetry-broken determinants, we obtain the correct ordering of energy levels with all the functionals even without multideterminant corrections. The corrections, however, improve the estimate of the excitation energy. In our view, one should generally apply multideterminant corrections, but in some cases their effect is small. This, for example, happens in the spin purification given by equation (5) when the singlet and triplet states are nearly degenerate. One such case is the 90 degree twisted ethylene molecule where the double bond has broken and the ground state is considered to be multiconfigurational but the singlet-triplet gap is so small that the broken spin symmetry solution gives a good estimate of the energy (see reference 45).

---

## Round 2 · Referee Report · Anonymous (Referee 2) · 2023-5-9

Report

The revisions are appropriate responses for Reviewer 1. I also agree with the authors that the journal they submitted their manuscript is appropriate.

---

## Round 2 · Referee Report · Anonymous (Referee 1) · 2023-5-15

Report

On the basis of the revisions made by the authors and their motivation letter, I think that the work is suitable for publication in Scipost Physics in its present form.

---

## Round 2 · Author Response

We hereby resubmit our manuscript to Scipost Physics and kindly ask that you to reconsider the redirection to SciPost Physics Core. Since SciPost Physics Core is not indexed in the Web of Science, it is of much less value for us, especially the early-stage co-authors of the manuscript.

Referee 2 does not question the suitability of our manuscript for publication in SciPost Physics, but referee 1 recommends publication in SciPost Physics Core. While we agree with referee 1 that our manuscript does not meet the first listed acceptance criterion for Scipost Physics (see https://scipost.org/SciPostPhys/about), i.e:
"Detail a groundbreaking theoretical/experimental/computational discovery",
we believe that the manuscript fulfills the third criterion:
"Open a new pathway in an existing or a new research direction, with clear potential for multipronged follow-up work".
Note that *only one* of the four listed acceptance criteria needs to be fulfilled for publication in SciPost Physics. The manuscript fulfills the latter criterion as it presents an approach that is much more efficient in terms of implementation and computational effort than previous methods, and yet has the predictive power of higher level approaches that are currently believed to be necessary to calculate excited electronic states in solids. In this way, our approach opens a new pathway for theoretical research on excited defect states in solids and will likely promote numerous studies, especially on large systems of interest for applications such as quantum computing. Such studies are currently limited because commonly used many-body approaches are too tedious and computationally intensive. The referees do not question the approach we present or our claims for its promise. We have already received requests from workers in the field who are interested in using our approach and it will soon be available in commonly used software.

---

## Round 2 · List of Changes

In response to question 1 of referee 1:
The following has been added at the end of section 2:  "In order to ensure the 511-atom supercell (with the vacancy) is large enough, calculations were also carried out with a smaller cell containing 215 atoms, and the results were found to differ by at most 5 meV (see Tables 2, and 3 in the Appendix). Given this small difference, it is appropriate to compare the results obtained here with a 511-atom supercell with the previous periodic calculations where a 215-atom supercell was used." 

In response to question 2 of referee 1:
We have added the following sentence at the end of the second paragraph of section 4: "We note that the commonly used self-consistent field optimization of orbitals typically converges on a symmetric solution if it is started from a symmetric initial wave function."

In response to question 4 of referee 1:
We have rephrased a bit the discussion in the second paragraph of the Results section to: "The correct ordering of the energy levels is obtained for all of the functionals. The largest absolute changes in excitation energy with respect to the density functional are obtained for the excited singlet state, 1A1, but the relative changes of the two singlet states are similar. The excited triplet state, which is the only state apart from the ground state that can be described using a single determinant, is affected less by the choice of functional.”

In response to question 6 of referee 1:
To clarify this, we have modified slightly a sentence in the last paragraph of section 2 to read "The energy of vertical excitations for both singlet and triplet states is obtained at the geometry optimized in the triplet ground state".

---

## Editorial Decision

published